# Online sexual health survey implementation: Practical lessons from a comparative, mixed-methods analysis of 30 countries

Hanna Saltis[1,2]*, Fiorella Farje[3,4], Simukai Shamu[5,6], Raquel Gomez[7], Johanna Schröder[8], Priya Kosana[9], Sharyn Burns[1,2], Jacqueline Hendriks[1,2], Olivia Williams[1,2], Kristien Michielsen[4,10], Joseph D. Tucker[11]*

**1** School of Population Health, Curtin University, Perth, Australia, **2** Curtin University, Collaboration for Evidence, Research and Impact in Public Health, Curtin University, Perth, Australia, **3** International Centre for Reproductive Health, Department of Public Health and Primary Care, Faculty of Medicine and Health Sciences, University of Ghent, Gent, Belgium, **4** Academic Network for Sexual and Reproductive Health and Rights Policy, Ghent University, Ghent, Belgium, **5** School of Public Health, University of the Witwatersrand, Johannesburg, South Africa, **6** Research Unit, Centre for Evaluation of Public Health Interventions in Africa (CEPHIA), Pretoria, South Africa, **7** Research Group Self-Regulation and Health, Institute for Health and Behaviour, Department of Behavioural and Cognitive Sciences, Faculty of Humanities, Education, and Social Sciences, University of Luxembourg - Campus Belval, Esch-sur-Alzette, Luxembourg, **8** Institute for Clinical Psychology and Psychotherapy, Department of Psychology, Medical School Hamburg, Hamburg, Germany, **9** Yale School of Public Health, Department of Epidemiology of Microbial Diseases, New Haven, Connecticut, United States of America, **10** Institute for Family and Sexuality Studies, Department of Neurosciences, Faculty of Medicine, KU Leuven, Leuven, Belgium, **11** School of Medicine, University of North Carolina, Chapel Hill, North Carolina, United States of America

* hanna.saltis@curtin.edu.au (HS); jdtucker@med.unc.edu (JT)

## Abstract

Online surveys are potentially useful for sexual and reproductive health. However, there are many persistent problems related to online sexual and reproductive health surveys. This study aims to understand online recruitment, dissemination and implementation as part of the International Sexual Health And REproductive health (I-SHARE) consortium online sexual health survey. This study used a mixed-methods, cross-sectional, multi-country design. We used survey data from the I-SHARE study and organised a separate implementation survey completed by I-SHARE country leads. A total of 24,004 participants in 30 countries responded to the I-SHARE survey. All countries implemented the I-SHARE survey online and most (n = 27, 90%) used convenience sampling. Social media promotion (n = 27, 90%), and partner organisations sharing (n = 21, 70%) were the most common recruitment methods. Twenty-nine countries responded to the implementation survey. We identified three themes related to online survey implementation: (1) Adaptation and flexibility highlighted research teams' responsiveness to rapidly changing contexts; (2) Better together: Partnerships illustrated the importance of multi-sectoral collaboration; and (3) Same but different: the heterogeneity of countries captured the ongoing tension between creating a standardised tool while honouring countries' unique socio-health climates and responses to the unfolding pandemic. This data demonstrates the

**Data availability statement:** All relevant data are available at https://dataverse.unc.edu/dataset.xhtml?persistentId=doi:10.15139/S3/5PZD6R.

**Funding:** The author(s) received no specific funding for this work.

**Competing interests:** None.

potential for using online sexual health surveys in diverse settings. Our study suggests the need for greater consideration of bias related to communication, especially the digital divide, when designing and implementing online surveys.

## Introduction

Online research implementation, dissemination and recruitment remain challenges for many researchers. The benefits and challenges of implementing online surveys are well-documented. They can reach large samples, are rapidly disseminated and can enhance anonymity [1–3]. They are cost-effective compared to in-person data collection methods [1–5]. They provide non-contact recruitment strategies for working with at-risk populations [3], and vulnerable groups [2]. However, online data is susceptible to sampling bias [1,2,6], lower generalisability [1–3,6,7], and the digital divide [8,9]. The digital divide refers to social inequities that prevent or limit access to the internet [8]. Differences in digital access may hinder equitable sexual health research and sampling rigour/integrity [2,6–8,10].

The efficacy of online methods such as social media in equitable research dissemination and recruitment is controversial. The debate is underpinned by data-sharing [3,11–13] and other privacy issues [3,11,13–18], and variable financial costs [12,13,17–21]. Social media give researchers access to a diverse range of potential participants, with 5.07 billion users worldwide [22]. However, some studies suggest that surveys disseminated online, particularly through social media, may not reach vulnerable groups such as older [13,23] and less educated [24] populations, thus widening the digital divide. Indeed, those who experience the burden of sexual health inequalities are often less digitally literate [25]. Using social media to recruit such specific vulnerable, or otherwise hard-to-reach populations has achieved mixed results, e.g., [2,13,17,26,27]. Conversely, other studies show that social media can be feasible [18,19], acceptable [18], effective [16], cost-effective [16,19], and accessible [19] recruitment tools, which potentially close digital gaps.

It is essential to understand the efficacy of online recruitment methods such as social media amongst specific demographics in particular geographical contexts. However, there remains a lack of evidence to demonstrate its effectiveness when recruiting broad samples in diverse, multi-country, sociocultural contexts. This study aims to address this gap, comparing the recruitment methods and numbers of 30 countries that disseminated a sexual health survey in 2020, during the first year of the COVID-19 pandemic, all online, and largely through social media.

The International Sexual Health and Reproductive Health (I-SHARE) Study was established to assess sexual health behaviours and services during this pandemic. The conglomerate was comprised of researchers from 30 High- (HIC), middle- (MIC) and low-income countries (LIC) [28,29]. Like most research during the pandemic, the I-SHARE Study was disseminated and implemented online. The survey was designed to capture a broad cross-section of the adult population. Eligibility criteria

were that respondents must be aged ≥18 years (though some countries included ethical provisions to enable younger people to participate), lived in a participating country, could read the survey language, could access the survey online and provide informed consent [28]. While this research was conducted in the COVID-19 context, the lessons learned may assist in helping recruit large, broad samples as well as vulnerable, hard-to-reach populations across a variety of socio-cultural and geopolitical contexts.

## Materials and methods

### Study design

This cross-sectional, multi-country study combined data from two sources. Data were sourced from the I-SHARE master data file. The original survey data was collected between 20 July 2020–15 February 2021 and examined participants' demographics, experiences or changes in sexual and reproductive health and associated behaviours during COVID-19 lockdowns [28,29]. Further information about the original study is available in the I-SHARE Study protocol paper [28]. The present study analysed participants' age, sex assigned at birth, and gender identity. In some countries, the variable *gender* was not included in the survey (see Table 1).

As part of the I-SHARE consortium, the implementation research team designed and distributed a separate survey for the 30 in-country leads involved in I-SHARE, which was disseminated between June and August 2021. This survey collected closed-ended responses about recruitment, dissemination, implementation, and sampling methods. In-country leads also responded to open text fields asking about the challenges and successes they faced and their likely causes. They were asked to provide two lessons they learned through their experiences of implementation and dissemination during their participation in I-SHARE and to add any additional relevant information. Responses were recorded on Google Forms.

### Analysis

**Quantitative analysis.** Due to the exploratory nature and small sample size of this study, there was insufficient power to conduct inferential statistical analyses. As such, descriptive statistics, including frequencies and percentages, were used to assess demographics and the methodologies employed by various countries and were calculated in SPSS. One country was counted twice as they employed two different sampling techniques (convenience and population-based sampling), so percentages may not add to 100%.

**Qualitative analysis.** Qualitative analysis of open-ended questions employed a phenomenological methodology, guided by social constructionist epistemology and an ecological model of health framework [30]. HS undertook the initial inductive, conventional content analysis [31–33]. Open coding was conducted on meaning units (units of analysis), which were defined here as short phrases or sentences. Codes were created and managed in NVIVO software. Categories were established from the codes [32], and themes were further developed from these categories, depicted below (see Fig 1). To ensure rigour and trustworthiness, the authorship team met to discuss codes, categories, and themes. Themes and interpretation were reviewed closely by HS and JT and further refined. The authorship group were also invited to further comment and refine themes throughout the editing process. To enhance rigour and reduce the risk of bias, the codes, categories and themes were triangulated with the quantitative data. HS also engaged in a reflexive journaling process while analysing the codes.

**Ethics.** All I-SHARE partners obtained ethics approval from their university's review board or ethics committee and signed a data-sharing agreement before inclusion in the multicounty analysis [28]. Participants in the study were given a detailed information form to read before commencing the survey and provided consent before commencing the survey [28,29]. Member-checking was carried out for the qualitative responses; each in-country lead whose quotes were used in this article gave written consent for these to be used. Quotes were de-identified on request.

**Table 1. Sociodemographic characteristics of the I-SHARE Study participants by country.**

| Country | N | Mean age | Sex | | | | | | Gender | | | | | | Area | | | |
|---|---|---|---|---|---|---|---|---|---|---|---|---|---|---|---|---|---|---|
| | | | Male | % | Female | % | Other | % | Cis-gen-der | % | non-cisgender | % | Other | % | (Semi-)rural | % | (Semi-)urban | % |
| Argentina | 845 | 34.8 | 166 | 19.6% | 679 | 80.4% | 0 | 0 | 841 | 99.5% | 4 | 0.5% | 0 | 0.0% | 105 | 12.6% | 728 | 87.4% |
| Australia | 561 | 33.2 | 149 | 26.6% | 408 | 72.7% | 4 | 0.007 | 515 | 91.8% | 38 | 6.8% | 8 | 1.4% | 55 | 9.9% | 503 | 90.1% |
| Botswana | 344 | 28.1 | 62 | 18.0% | 281 | 81.7% | 1 | 0.003 | 339 | 98.5% | 4 | 1.2% | 1 | 0.3% | 111 | 32.9% | 226 | 67.1% |
| Canada | 163 | 34.3 | 24 | 15.2% | 133 | 84.2% | 1 | 0.006 | 149 | 95.5% | 1 | 0.6% | 6 | 3.8% | 25 | 16.1% | 130 | 83.9% |
| China | 827 | 27.8 | 416 | 50.3% | 391 | 47.3% | 20 | 0.024 | | | | | | | 171 | 21.1% | 638 | 78.9% |
| Colombia | 2452 | 30.9 | 945 | 38.5% | 1505 | 61.4% | 2 | 0.001 | 2409 | 98.2% | 43 | 1.8% | 0 | 0.0% | 32 | 1.3% | 2420 | 98.7% |
| Czech Republic | 1862 | 42.8 | 850 | 45.6% | 1009 | 54.2% | 3 | 0.002 | 1740 | 93.8% | 114 | 6.1% | 2 | 0.1% | 1035 | 56.2% | 806 | 43.8% |
| Denmark | 1001 | 51.3 | 459 | 45.9% | 542 | 54.1% | 0 | 0 | 983 | 99.1% | 9 | 0.9% | 0 | 0.0% | 438 | 44.2% | 552 | 55.8% |
| Egypt | 968 | 31.9 | 509 | 52.6% | 450 | 46.5% | 9 | 0.009 | | | | | | | 370 | 38.2% | 598 | 61.8% |
| France | 1593 | 30.9 | 277 | 17.4% | 1314 | 82.5% | 2 | 0.001 | 1546 | 97.2% | 11 | 0.7% | 33 | 2.1% | 500 | 31.4% | 1092 | 68.6% |
| Germany | 612 | 28 | 106 | 17.3% | 505 | 82.5% | 1 | 0.002 | 581 | 95.1% | 25 | 4.1% | 5 | 0.8% | 122 | 20.1% | 486 | 79.9% |
| Italy | 329 | 34.2 | 113 | 34.3% | 216 | 65.7% | 0 | 0 | 311 | 94.8% | 17 | 5.2% | 0 | 0.0% | 104 | 31.6% | 225 | 68.4% |
| Kenya | 243 | | 80 | 32.9% | 162 | 66.7% | 1 | 0.004 | 239 | 99.6% | 1 | 0.4% | 0 | 0.0% | | | | |
| Latvia | 1176 | 32.4 | 204 | 17.3% | 969 | 82.4% | 3 | 0.003 | | | | | | | 126 | 10.8% | 1037 | 89.2% |
| Lebanon | 54 | 31 | 44 | 81.5% | 10 | 18.5% | 0 | 0 | 50 | 96.2% | 2 | 3.8% | 0 | 0.0% | 7 | 13.5% | 45 | 86.5% |
| Luxembourg | 568 | 35.8 | | | | | | | | | | | | | 295 | 53.1% | 261 | 46.9% |
| Malaysia | 499 | 32.9 | 276 | 55.3% | 223 | 44.7% | 0 | 0 | | | | | | | 102 | 20.9% | 385 | 79.1% |
| Mexico | 1673 | 38.5 | 392 | 23.4% | 1281 | 76.6% | 0 | 0 | 1581 | 94.9% | 81 | 4.9% | 4 | 0.2% | 157 | 9.4% | 1505 | 90.6% |
| Moldova | 244 | 33.7 | 47 | 19.3% | 196 | 80.3% | 1 | 0.004 | 236 | 97.1% | 7 | 2.9% | 0 | 0.0% | 26 | 10.7% | 218 | 89.3% |
| Mozambique | 66 | 33.7 | 27 | 40.9% | 39 | 59.1% | 0 | 0 | 60 | 92.3% | 5 | 7.7% | 0 | 0.0% | 3 | 4.7% | 61 | 95.3% |
| Nigeria | 231 | 26.7 | 100 | 43.3% | 131 | 56.7% | 0 | 0 | 228 | 98.7% | 3 | 1.3% | 0 | 0.0% | 37 | 16.1% | 193 | 83.9% |
| Panama | 960 | 31.3 | 397 | 41.4% | 563 | 58.6% | 0 | 0 | 884 | 92.9% | 66 | 6.9% | 2 | 0.2% | 194 | 20.3% | 764 | 79.7% |
| Portugal | 3323 | 32.4 | 618 | 18.6% | 2702 | 81.3% | 3 | 0.001 | 3086 | 93.2% | 216 | 6.5% | 10 | 0.3% | 840 | 25.5% | 2459 | 74.5% |
| Singapore | 566 | 28.8 | 312 | 55.1% | 254 | 44.9% | 0 | 0 | 525 | 93.4% | 37 | 6.6% | 0 | 0.0% | 0 | 0.0% | 566 | 100.0% |
| South Africa | 29 | 35.9 | 17 | 58.6% | 12 | 41.4% | 0 | 0 | 28 | 100.0% | 0 | 0.0% | 0 | 0.0% | 4 | 14.3% | 24 | 85.7% |
| Spain | 295 | 34.1 | 68 | 23.1% | 227 | 76.9% | 0 | 0 | 286 | 97.6% | 4 | 1.4% | 3 | 1.0% | 105 | 35.7% | 189 | 64.3% |
| Sweden | 1307 | 34.1 | 657 | 50.3% | 644 | 49.3% | 6 | 0.005 | 1220 | 93.3% | 80 | 6.1% | 7 | 0.5% | | | | |
| Uganda | 212 | 34 | 101 | 47.6% | 108 | 50.9% | 3 | 0.014 | | | | | | | 31 | 15.0% | 175 | 85.0% |
| Uruguay | 696 | 35.4 | 198 | 28.4% | 497 | 71.4% | 1 | 0.001 | 678 | 97.8% | 13 | 1.9% | 2 | 0.3% | 47 | 6.9% | 639 | 93.1% |
| USA | 305 | 37.4 | 72 | 24.2% | 226 | 75.8% | 0 | 0 | 284 | 95.6% | 4 | 1.3% | 9 | 3.0% | 77 | 25.8% | 221 | 74.2% |
| **Total** | **24004** | **32,5** | **7686** | **32.8%** | **15677** | **66.9%** | **61** | **0.003** | **18799** | **95.5%** | **785** | **4.0%** | **92** | **0.5%** | **5119** | **23.0%** | **17146** | **77.0%** |

Note: I-SHARE = the International Sexual Health And REproductive health.

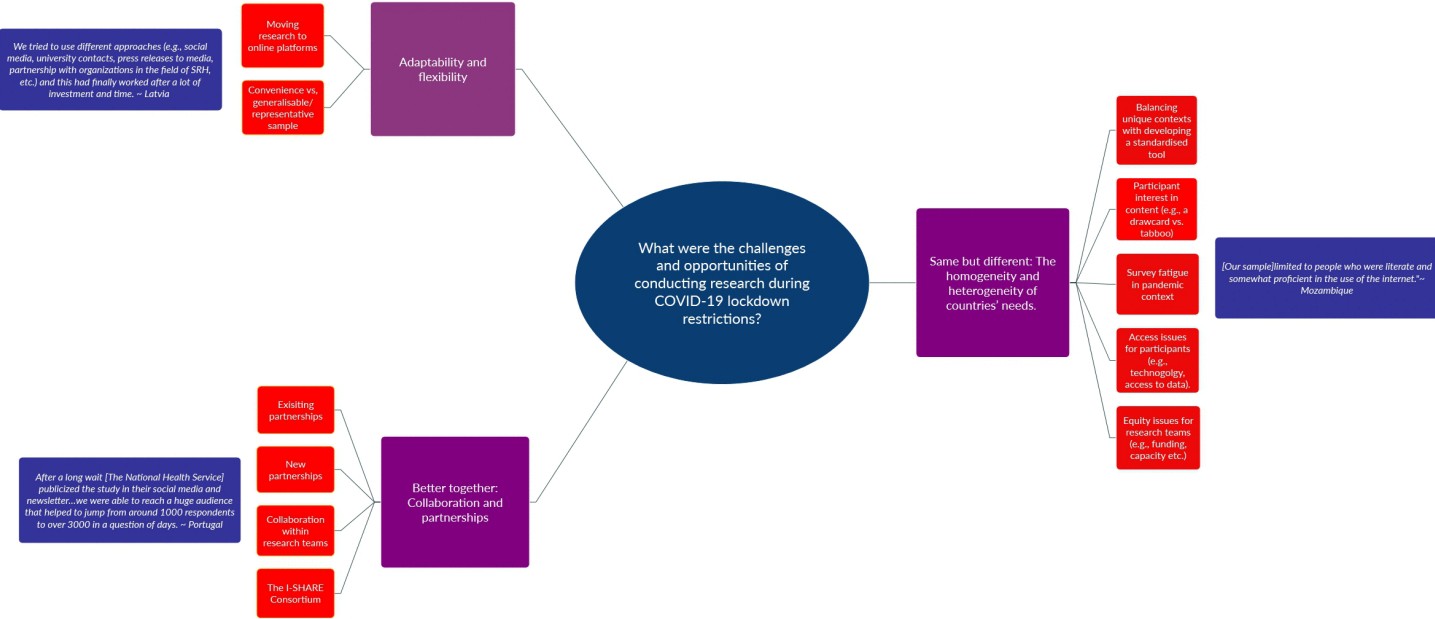

**Fig 1. Thematic map of content analysis.** Note: Purple squares represent themes, orange boxes represent categories, and blue boxes contain example quotes.

## Results

### Quantitative results

**Participation.** In total, 24,004 participants across 30 countries responded to the I-SHARE survey. The mean age of the respondents was 32.5 years, and most were female (66.9%, sex assigned at birth). The majority lived in (semi-)urban areas (77.0%). Demographic data were missing for some countries (see Table 1). All 30 in-country leads who were involved in the first wave of the I-SHARE study completed the implementation survey.

**Implementation methods.** All countries implemented the I-SHARE survey online. Most countries employed convenience sampling (n = 28, 93.3%), and others used representative population-based sampling methods (n = 3, 10%). Four countries used incentives, including SGD$10 for all participants (Singapore), ₦200 phone cards (Nigeria), weekly draw prizes to win one of two AUD$50 gift cards (Australia), (10EGP) (Egypt) (see Table 2). Given the disparity between their participation numbers, and the type of incentive, a relationship between incentives and participation numbers cannot be established.

**Dissemination methods.** The most common recruitment method was promotion on social media (n = 27, 90%), followed by asking partner organisations to share the survey link (n = 21, 70%). Several countries used paid advertising on social media (n = 13, 43.3%). Twenty-six (90%) countries used Facebook, 19 (66%) used Instagram, 17 (59%) used WhatsApp, 15 (52%) used Twitter, and six (20%) used alternative social media platforms such as TikTok, LinkedIn, Reddit and We Chat (see Table 3).

Research teams that recruited fewer than 200 participants typically used only two recruitment methods. Those that recruited between 200 and 499 participants used 3–4 recruitment methods; the number of recruitment methods varied for countries with more than 500 participants. The team in Luxembourg used six methods, while the teams in Latvia (n = 1176) and Colombia (n = 2452) used five methods, Portugal's team (n = 3323) used four methods while China (n = 827) and Denmark (n = 1001), used only one method (see Table 3).

**Table 2. Participating I-SHARE countries' dissemination methods by sample size.**

| Nº | Country | Sample size | N of methods | Social media | Partner organisations shared | Paid social media advertising (e.g., Facebook or Instagram ads) | University Website | Other* | Over the phone | Unpaid traditional advertising (television or local media including newspapers etc.) | Did you provide incentives for your participants? | Research recruitment agency panel |
|---|---|---|---|---|---|---|---|---|---|---|---|---|
| n < 200 | South Africa | 29 | 2 | ● | ● | | | | | | | |
| | Lebanon | 54 | 2 | ● | ● | | | | | | | |
| | Mozambique | 66 | 3 | ● | ● | ● | | | | | | |
| | Canada | 163 | 2 | ● | ● | | | | | | | |
| n > 200 and < 500 | Uganda | 212 | 3 | ● | ● | | | ● | | | | |
| | Nigeria | 231 | 5 | ● | ● | | | ● | ● | | ● | |
| | Kenya | 243 | 3 | ● | | ● | | | ● | | | |
| | Moldova | 244 | 3 | ● | | | ● | | ● | | | |
| | Spain | 295 | 3 | ● | ● | | ● | | | | | |
| | USA | 305 | 2 | ● | ● | | | | | | | |
| | Italy | 329 | 3 | ● | ● | ● | | | | | | |
| | Botswana | 344 | 4 | ● | ● | ● | | ● | | | | |
| | Malaysia | 499 | 3 | ● | ● | | ● | | | | | |
| n > 500 | Australia | 561 | 5 | ● | ● | ● | ● | | | | ● | |
| | Singapore | 566 | 3 | ● | | ● | ● | | | | | |
| | Luxembourg | 568 | 6 | ● | ● | ● | ● | ● | | ● | | |
| | Germany | 612 | 3 | ● | ● | | ● | | | | | |
| | Uruguay | 696 | 3 | ● | ● | ● | | | | | | |
| | China | 827 | 1 | ● | | | | | | | | |
| | Argentina | 845 | 2 | ● | | | | ● | | | | |
| | Panama | 960 | 3 | ● | | ● | ● | | | | | |
| | Egypt | 968 | 3 | ● | ● | | | | | | ● | |
| | Denmark | 1001 | 1 | | | | | | | | | ● |
| | Latvia | 1176 | 5 | ● | ● | | ● | ● | | ● | | |
| | Sweden | 1307 | 2 | | | | ● | | | | | ● |
| | France | 1593 | 3 | ● | ● | ● | | | | | | |
| | Mexico | 1673 | 4 | ● | ● | ● | | ● | | | | |
| | Czech Republic | 1862 | 3 | | | | ● | | | | ● | ● |
| | Colombia | 2452 | 5 | ● | ● | ● | | | ● | ● | | |
| | Portugal | 3323 | 4 | ● | ● | ● | | | | | | |
| | **Total** | **24004** | | **27** | **21** | **13** | **11** | **7** | **4** | **4** | **4** | **3** |
| | | | | **90.0%** | **70.0%** | **43.3%** | **36.6%** | **23.3%** | **13.3%** | **13.3%** | **13.3%** | **10.0%** |

*Other methods include institutional mailing list/email, one-on-one discussions with eligible people in their networks and shared with staff in related organizations, posters, institutional website, and invitation leaflet distribution in antenatal clinics, urological departments, HIV clinic.

**Table 3. Social media platforms used by participating I-SHARE country and level of recruitment.**

| Nº | Country | Sample size | N social media platform | Face-book | Insta-gram | WhatsApp | Twitter | Reddit | LinkedIn | TikTok | We Chat |
|---|---|---|---|---|---|---|---|---|---|---|---|
| en <200 | South Africa | 29 | 4 | ✓ | ✓ | ✓ | ✓ | | | | |
| | Lebanon | 54 | 3 | | ✓ | ✓ | ✓ | | | | |
| | Mozambique | 66 | 2 | ✓ | | ✓ | | | | | |
| | Canada | 163 | 3 | ✓ | ✓ | | ✓ | | | | |
| n>200 and <500 | Uganda | 212 | 2 | ✓ | | ✓ | | | | | |
| | Nigeria | 231 | 5 | ✓ | ✓ | ✓ | ✓ | | ✓ | | |
| | Kenya | 243 | 3 | ✓ | | ✓ | ✓ | | | | |
| | Moldova | 244 | 2 | ✓ | ✓ | | | | | | |
| | Spain | 295 | 4 | ✓ | ✓ | ✓ | ✓ | | | | |
| | USA | 305 | 6 | ✓ | ✓ | ✓ | ✓ | | ✓ | ✓ | |
| | Italy | 329 | 4 | ✓ | ✓ | | ✓ | | ✓ | | |
| | Botswana | 344 | 3 | ✓ | ✓ | ✓ | | | | | |
| | Malaysia | 499 | 5 | ✓ | ✓ | ✓ | ✓ | ✓ | | | |
| n<500 | Australia | 561 | 4 | ✓ | ✓ | | ✓ | ✓ | | | |
| | Singapore | 566 | 1 | ✓ | | | | | | | |
| | Luxembourg | 568 | 1 | ✓ | | | | | | | |
| | Germany | 612 | 3 | ✓ | ✓ | | ✓ | | | | |
| | Uruguay | 696 | 4 | ✓ | ✓ | ✓ | ✓ | | | | |
| | China | 827 | 1 | | | | | | | | ✓ |
| | Argentina | 845 | 3 | ✓ | ✓ | ✓ | | | | | |
| | Panama | 960 | 4 | ✓ | ✓ | ✓ | ✓ | | | | |
| | Egypt | 968 | 1 | ✓ | | | | | | | |
| | Denmark | 1001 | 0 | | | | | | | | |
| | Latvia | 1176 | 3 | ✓ | ✓ | | ✓ | | | | |
| | Sweden | 1307 | 0 | | | | | | | | |
| | France | 1593 | 2 | ✓ | ✓ | | | | | | |
| | Mexico | 1673 | 3 | ✓ | | ✓ | ✓ | | | | |
| | Czech Republic | 1862 | 1 | ✓ | | | | | | | |
| | Colombia | 2452 | 4 | ✓ | ✓ | ✓ | ✓ | | | | |
| | Portugal | 3323 | 3 | ✓ | ✓ | ✓ | | | | | |
| Total | | 23004 | | 27 | 19 | 17 | 15 | 3 | 2 | 1 | 1 |
| | | | | 86.7% | 63.3% | 56.7% | 50.0% | 10.0% | 6.7% | 3.3% | 3.3% |

## Qualitative analysis

One country was excluded from the qualitative analysis, as they replied "N/A" to all open-ended questions. The country which used two different sampling techniques (convenience and population-based sampling) reported different challenges and successes, respectively, and was counted twice.

**Content analysis.** Twenty-nine of the in-country leads responded to the open-ended questions in the survey and identified a range of challenges, successes, and lessons learned about online research dissemination and implementation throughout wave one of the I-SHARE Study. Three themes were derived from the data: (1) Adaptation and flexibility, (2) Better together: Collaboration and partnerships, and (3) Same but different: the homogeneity and heterogeneity of countries. A thematic map is provided below (see Fig 1).

Notably, two in-country leads who used population-based samples reported no challenges. One attributed this directly to using this sampling technique, stating, "There were not any [challenges] because, for the population-based sample, we used the research agency panel". Conversely, three countries reported not reaching the minimum participation level (n = 200) for inclusion in a larger analysis. Possible reasons for these low numbers will be explored in theme three.

**Adaptability and flexibility.** Most in-country leads discussed how online platforms allowed innovative adaptation to facilitate research dissemination and implementation. For example, Colombia's lead stated, "it is possible to apply complex online surveys with high-quality standards." Similarly, the Australian in-country lead noted that online surveys created easier, more targeted dissemination, "[paid] social media ads were easily disseminated. We were able to target different population groups." The South African lead noted that "online platforms could be helpful for research during challenging times requiring social distancing." Together, these examples highlight the utility of online surveys, particularly in a global context, as they are flexible and adaptable for diverse populations and global research teams within a rapidly evolving, unpredictable climate, without compromising data quality.

Despite these advantages, some countries' leads expressed concerns about sample representativeness. Uruguay's lead commented, "We could not…capture a sample that approximated the profiles of the general population...we achieved a sample of 673 participants." Here, a compromise between rapid data collection from a large sample and a generalisable sample is evident. Botswana and Australia attempted to counteract demographic imbalances through paid, targeted social media advertising. Botswana's in-country lead illustrated the importance of financial support in assisting in this type of dissemination, saying, "We got just over 400 responses in Botswana, the majority through [paid] social media promotion. Social media promotion without sponsored posts was not very successful..." Argentina's lead said, "it would have been better if we could have paid

for a dissemination and tried to get a representative sample." The in-country lead for the Czech Republic said, "our success is that we got a financial grant for the population-based sample, that made it much easier for us." Thus, funding when recruiting online is essential as paid advertising increases opportunities to achieve a representative sample and target harder-to-reach populations. However, accessing funding may not always be achievable, particularly when data must be collected rapidly.

In-country leads with high response rates supplemented demographic deficits by using multiple methods. Portugal's in-country lead highlights, "We tried to use different approaches (e.g., social media, university contacts, press releases to media, partnership with organizations in the field of sexual and reproductive health, etc.) and this had finally worked after a lot of investment and time." Contrastingly, other high-recruiting countries used only one technique with a comparable effect. For example, Denmark's in-country lead stated: "It was the first time our research group tried having a sample drawn and sending invitations through E-boks (a secure, online, Danish digital communications server that organisations can utilise to reach large groups). …this is an incredibly efficient way to recruit participants and we are currently using it in another study." This flexibility and willingness to adapt to unprecedented circumstances can produce enduring, positive changes in research processes. Moreover, these examples highlight that either triangulating multiple methods or a dedicated singular method can be effective in off-setting sampling biases and countries may choose which is more suitable depending on their individual needs and resources.

**Better together: Collaboration and partnerships.** Many in-country leads discussed the importance of partnerships and collaborations for successful recruitment and dissemination. Most mobilised their existing partnerships. Panama's lead noted, "Due to this [pushback from conservatives], we moved away from institutional sharing to I-SHARE-Panama team member sharing and networked across the country this way." Having existing external partnerships with people who support the research area was thus important in recruiting in the face of adversity.

Some countries, like Portugal's team, established new key networks and reported success in having a large media company disseminate their research, "In more than one major newspaper - mostly using their social media platforms", Portugal's in-country lead also collaborated with their National Health Service, stating,

After a long wait, [the National Health Service] publicized the study in their social media and newsletter. …since their Facebook was one of the most seen in Portugal – the first place where the daily number of COVID cases were reported – we were able to reach a huge audience that helped to jump from around 1000 respondents to over 3000 in a question of days.

Findings from Portugal's team suggest that government endorsement of sexual and reproductive health research may facilitate successful recruitment.

Similarly, France's lead stated, "Few responses with Facebook and Instagram alone. We also struggled to recruit men. We were successful in recruiting when the link was shared by partner organisations (planned parenthood) or influential stockholders on Facebook." Germany's lead, reiterated this, saying "One popular feminist Instagram account posted the link. This resulted in a larger response in the end." Taken together, it suggests that utilising pre-existing, large networks may be more useful than social media advertising.

Multiple collaborations were essential in recruiting larger participant numbers. Latvia recruited using social media and via more traditional media like television, and traditional methods such as advertisements, and invitation leaflets distributed in antenatal clinics, urological departments, and clinics for people living with HIV. Collaboration within the research team was also essential in driving recruitment. Latvia's lead lauded their inclusion of young researchers, stating, "we involved young scientists – students and an obstetrics/gynaecology resident – it helped a lot for using social media, promoting the I-SHARE among young people and later during the qualitative research part." The ability to diversify the research team expanded their skillset and, by extension, their recruitment methods. This, in conjunction with the use of multiple methods, ultimately led to greater recruitment numbers for Latvia.

Several teams acknowledged the support of the I-SHARE consortium as core to their positive experience, regardless of their participation numbers. As the lead for South Africa stated, "I must say - here in South Africa, we were not [as] successful as we wished, but obviously, the support received from the [I-SHARE] team was very instrumental. I greatly liked the collaborative efforts that brought colleagues from different parts of the world to make the study a reality. I believe it is an opportunity for long-term collaborations." This demonstrates that partnership and collaboration amongst researchers on a global level provided a crucial support system during difficult times, as well as an opportunity for ongoing collaborations and relationships beyond crisis contexts.

**Same but different: The homogeneity and heterogeneity of countries' needs.** This theme captures the ongoing tension between creating a standardised tool while honouring countries' unique socio-health climates and responses to the unfolding pandemic (see Fig 2). As the in-country lead for Italy noted, "Multicultural surveys are very engaging but also difficult. Variables to consider are enormous, and challenges may vary from country to country".

The leads from Egypt, Italy, and Malaysia discussed the content of the survey as a potential challenge for recruitment. Egypt's in-country lead illustrates this, stating, "the survey is long and about sensitive issues as regards the Egyptian culture." This highlights how reproductive and sexual health may be taboo and pose a cultural barrier to recruitment in some countries or regions.

In-country leads speculated about the impacts of their distinct pandemic circumstances on dissemination and implementation. The in-country lead for South Africa reflected on the difficulty of standing out amongst the proliferation of survey research, stating, "by the time we sent the survey out, the population had fatigue for COVID surveys, and the lockdown levels had changed a couple of times". This captures the difficulties for researchers in responding in real-time to rapidly changing conditions and how such fluctuating conditions contributed to a sense of civil exhaustion, exacerbated by constant online engagement.

Similarly, in Australia, during the survey dissemination, COVID-19 cases and isolation restrictions varied considerably between States and Territories. Australia's lead considered, "Recruitment may be easier when we are in a specific lockdown." This suggests that survey uptake may depend on the perceived relevance and societal notions of COVID-19 risk

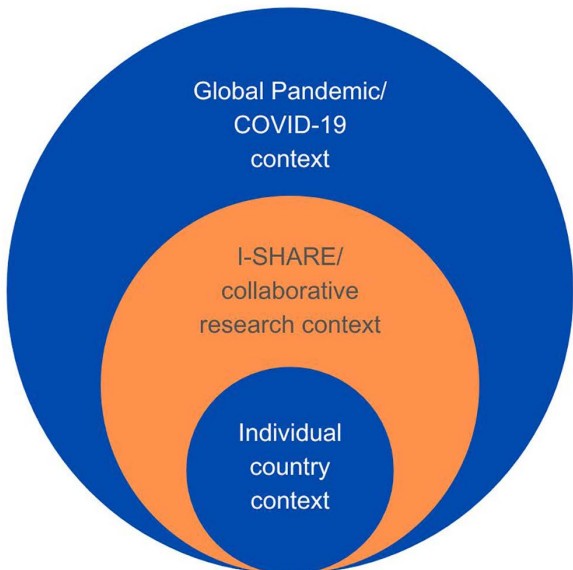

**Fig 2. A conceptual diagram of interconnected contexts affecting data collection during the I-SHARE study.** The contexts are embedded to demonstrate that each is intrinsically linked to and influences the other spheres.

and attitudes at different stages of the pandemic, which are likely influenced by infection rates and experiences of isolation measures. Likewise, the in-country lead for Lebanon reflected, "COVID-19 was a big challenge, and the economic and humanitarian crisis in the country made it impossible for people to get attracted to the survey." This indicates that online survey completion is a low priority in comparison to navigating financial and other socio-ecological factors that would become more salient to individuals in certain contexts.

Inequities, inequalities, and barriers to survey access were observed in several countries. For example, Colombia's in-country lead noted that "the recruitment of populations aged 35 + -- this population is less prone to use technology for giving personal information through this channel". Crucially, this suggests that potentially, the voices of vulnerable populations may not be captured. This was reiterated by several other countries, including Mozambique, whose lead expressed concern that their sample was "limited to people who were literate and somewhat proficient in the use of the internet." Discussion of these sampling challenges highlights existing inequities and invites researchers to reflect on strategies to include those invisible, vulnerable, and likely in need of the most aid to participate in online research.

## Discussion

This study collates quantitative and qualitative data highlighting challenges and successes in using online research methods. Online survey design needs to consider the digital divide. Our data align with previous research highlighting the mixed utility of online surveys in reaching specific populations [1,3].

Most countries used online convenience sampling. The challenges that in-country leads named are consistent with research that suggests online convenience samples are less representative than other sampling methods [4,6]. In contrast, several research teams used online population-based sampling. These samples have greater generalizability compared to convenience sampling. The Natsal-COVID study in the UK achieved a quasi-representative sample using an online approach. Online population-representative panels are costly, requiring greater resources [1,3,6],

Our data suggest that online surveys may over-sample women compared to men. Paid social media may help to increase the representation of certain demographics in online samples (e.g., men) [20]. This is consistent with research

on online recruitment strategies [e.g., 2,3,6,20,24]. Our findings suggest the need for communications expertise when designing online surveys [13].

This study highlighted that partnering with organisations and individuals who have large, followings may assist with recruitment. Previous research suggests that paid social media advertising may have low conversion rates when comparing reach to click-throughs to complete responses [18,20,21], and conversion costs can be more expensive than using influencers [21]. Our data suggest that partnerships with a trusted source, rather than advertisements, helped to address these challenges. This supports previous suggestions that involving key influencers and networks could enhance social media recruitment [21,34] compared to paid advertising [21].

This study supports evidence that utilising multiple engagement methods [13,35] can partially address the digital divide and increasing the likelihood of reaching under-served populations. Many vulnerable groups have poor access to the internet [25]. Alternative offline recruitment strategies are also needed to ensure researchers are reaching a range of vulnerable populations who may not be using social media.

For some countries, the taboo nature of sexual and reproductive health was perceived to impact recruitment. However, online surveys can enhance anonymity [1–3]. Researchers may want to highlight the increased privacy that surveys offer during recruitment to help allay potential participants' fears of discussing sensitive content and encourage participation. Where culturally appropriate, having trusted sources such as government websites (like Portugal), partner organisations, traditional media (i.e., television and newspapers) share sexual health surveys, may help to normalize these discussions and further encourage participation.

Consortia act as a support network, consultancy, and sounding board [36–38], and the collaborative approach of the I-SHARE network enabled discussions of shared and country-specific challenges and successes. Each team involved in the I-SHARE conglomerate was impacted by a complex combination of individual contextual factors, unique country environments, and the interests of the I-SHARE consortium (see Fig 2). The use of online meetings through Zoom allowed partnerships, collaboration, and problem-solving between diverse research teams from various countries and time zones. While it can be challenging to balance individual countries' interests and the collective interest of members in the consortium [10], the I-SHARE consortium demonstrated that international collaboration based on open science principles is possible across diverse countries.

In-country leads speculated about specific factors that impacted their unique experience, for example, easing restrictions at the time of dissemination, surviving COVID-19, and survey fatigue. Similar barriers are identified in previous research [2,3,6,26,27,37]. Unique societal stressors and competing interests may make online surveys, especially those related to sexual health, feel less relevant [26]. Research teams should consider highlighting the importance of sexual and reproductive health to overall well-being when designing online surveys. Ensuring that surveys are not onerous or lengthy, and providing financial incentives, may also reduce potential survey fatigue, though these considerations should be balanced with the potential for increased risk of survey fraud (especially when providing incentives) and the ability to capture complex data [3].

The findings from this study suggest that to effectively utilise online surveys and social media for research dissemination and recruitment, researchers should have a clear plan for how to utilise resources effectively. If funds are available, considering the use of an online panel may be the most effective way to gain a representative sample through online means. If this is unavailable, researchers are encouraged to understand the needs of their target demographics and how best to reach them. Where possible, include consultation with social media experts to help with creating effective advertising content, and to help with increasing the value of paid advertising (i.e., conversions from reach to completed surveys). When targeting specific populations, using partner organisations and key influencers to share the research is an effective way to reach large groups, and to save on advertising costs. To ensure that the research is not contributing to the digital divide, consider triangulation of online advertising with more traditional methods of data collection and recruitment. Lastly, consider how the current socio-cultural and political climates might influence response rates and factor this into the dissemination and recruitment strategies.

## Strengths and limitations

This study has several strengths, including the rigorous methodology of the I-SHARE study, and using data collected during the unfolding COVID-19 pandemic from multiple countries, and research teams. This study expands the literature by focusing on implementation and dissemination strategies from multiple, diverse countries that all used a similar survey instrument. Despite these considerable strengths, this study has some limitations.

First, the approach focused on the number of participants recruited as an indicator of effective implementation. In addition, to be eligible for inclusion in the ISHARE study, countries were required to meet the threshold of 200 participants, rather than a proportion of their population, and were not required to have a representative sample. However, there are few alternative metrics to gauge the relative effectiveness of surveys. Future studies should consider their potential pool of participants, and eligibility for inclusion should be based on a proportion of their national adult population. Second, the survey relied on self-reports from the primary in-country, chief investigators, and thus this study solely captures these individuals' perspectives, though others on their research teams may have been well-positioned to comment on their implementation and dissemination strategies. Third, inferential statistical analysis was beyond the scope of this study. Future research should consider inferential analysis that accounts for contextual variables highlighted in this study. Finally, all teams that participated were able to join the consortium despite substantial competing COVID-19 and non-COVID-related demands. The authors speculate that consortium teams may have greater access to resources than other teams who did not join. Notwithstanding these limitations, the present study provided a novel examination of the utility of online surveys and social media dissemination and implementation strategies appropriate in diverse contexts.

## Conclusions

This study shows how the utility of online surveys and social media recruitment and dissemination are responsive to sociocultural contexts, cultural events, and social consciousness. These methods were effective for recruiting larger numbers when done strategically. However, they also presented challenges for representativeness and access. Research teams in the I-SHARE network reported the positive influence of partnerships and collaboration and the support of the I-SHARE consortium in garnering success in this challenging climate. These methods were feasible in diverse settings, including many resource-constrained nations.

## Acknowledgments

We would like to thank Gabriella Perotta, Roccio Murad and the broader I-SHARE consortium for their guidance and assistance with this research.

## Author contributions

**Conceptualization:** Hanna Saltis, Fiorella Farje, Simukai Shamu, Raquel Gomez, Priya Kosana.

**Formal analysis:** Hanna Saltis, Fiorella Farje, Simukai Shamu, Johanna Schröder, Priya Kosana, Kristien Michielsen, Joseph D. Tucker.

**Investigation:** Joseph D. Tucker.

**Methodology:** Hanna Saltis, Fiorella Farje, Simukai Shamu, Raquel Gomez, Johanna Schröder, Kristien Michielsen, Joseph D. Tucker.

**Project administration:** Hanna Saltis, Joseph D. Tucker.

**Supervision:** Simukai Shamu, Sharyn Burns, Jacqueline Hendriks, Kristien Michielsen, Joseph D. Tucker.

**Writing – original draft:** Hanna Saltis, Fiorella Farje, Simukai Shamu, Raquel Gomez, Johanna Schröder, Joseph D. Tucker.

**Writing – review & editing:** Hanna Saltis, Fiorella Farje, Simukai Shamu, Raquel Gomez, Johanna Schröder, Priya Kosana, Sharyn Burns, Jacqueline Hendriks, Olivia Williams, Kristien Michielsen, Joseph D. Tucker.

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
