## [Decision Letter · Decision Letter 0]

27 Aug 2025

PONE-D-25-14440Online sexual health survey implementation: Practical lessons from a comparative, mixed-methods analysis of 30 countriesPLOS ONE

Dear Dr. Tucker,

Thank you for submitting your manuscript to PLOS ONE. After careful consideration, we feel that it has merit but does not fully meet PLOS ONE’s publication criteria as it currently stands. Therefore, we invite you to submit a revised version of the manuscript that addresses the points raised during the review process.

We look forward to receiving your revised manuscript.

Kind regards,

Ivan Alejandro Pulido Tarquino, MSc

Academic Editor

PLOS ONE

Journal Requirements:

2. Thank you for stating the following in your Competing Interests section: “None”

4. In this instance it seems there may be acceptable restrictions in place that prevent the public sharing of your minimal data. However, in line with our goal of ensuring long-term data availability to all interested researchers, PLOS’ Data Policy states that authors cannot be the sole named individuals responsible for ensuring data access (http://journals.plos.org/plosone/s/data-availability#loc-acceptable-data-sharing-methods).

**Additional Editor Comments:**

In particular, please address the following key points:

Abstract: Revise it as per review suggestion, you would resume the implications of your study findings.

Methods: Provide more detail on the process, including how many persons performed the analysis (simultaneously or independently), and how differing interpretations were resolved, highlighting what measures do you implement to maintain trustworthiness of your study findings.

Study population: Clarify the number of persons eligible per country, noting the large differences as mentioned by the reviewer.

Discussion: this section would provide comparisons of your results with other studies, especially those on sexual health, and reflect on the gender imbalance in responses (notably the higher participation of females).

Figures and tables: Improve the overall quality of figures and tables to ensure readability.

We believe these revisions will strengthen the clarity, quality, and international relevance of your manuscript. Please prepare and submit a revised version, together with a detailed response letter explaining how you have addressed each point.

We look forward to receiving your revised manuscript.

Thank you

Best regards

Ivan Alejandro Pulido Tarquino

Academic editor

Reviewers' comments:

Reviewer's Responses to Questions

**Comments to the Author**

1. Is the manuscript technically sound, and do the data support the conclusions?

Reviewer #1: Yes

2. Has the statistical analysis been performed appropriately and rigorously? 

Reviewer #1: N/A

3. Have the authors made all data underlying the findings in their manuscript fully available?

Reviewer #1: Yes

4. Is the manuscript presented in an intelligible fashion and written in standard English?

Reviewer #1: Yes

5. Review Comments to the Author

Reviewer #1: This is a very valuable study which is worth publication in a broadly available international journal.

But it needs some revisions: instead of last sentence in the abstract you should give some hints from your recommendations.

Content analysis should be described with more details: how many persons performed these (in parallel?) and how did you handle differing interpretations?

You should also mention the number of persons eligible per country since they are quite different (for example: China vs Luxembourg). Discussion is very lengthy and should not replicate too many details from results. Instead, you should draw comparisons to other studies with the theme of sexual health. Can you- in this context- come back to the fact that in most countries much more females than males answered to the survey?

I cannot read Figure 1- reproduction is bad

minor remarks:

line 84: you should write it's or the instead of it

line 94/95: What do you want to say with: The survey was designed to capture a broad cross-section of the adult population 15. which age range is meant?

6. PLOS authors have the option to publish the peer review history of their article (what does this mean?). If published, this will include your full peer review and any attached files.

Reviewer #1: **Yes:**Prof. Dr. med. Erika Baum

---

## [Author Response · Author response to Decision Letter 1]

3 Oct 2025

Dear Ivan Alejandro Pulido Tarquino,

See the attached file on responses to reviewer comments for a detailed description of responses and what has been changed.

Sincerely,

Hanna and Joe on behalf of the author team

---

## [Decision Letter · Decision Letter 1]

24 Nov 2025

PONE-D-25-14440R1Online sexual health survey implementation: Practical lessons from a comparative, mixed-methods analysis of 30 countriesPLOS ONE

Dear Dr. Tucker,

Thank you for submitting your manuscript to PLOS ONE. After careful consideration, we feel that it has merit but does not fully meet PLOS ONE’s publication criteria as it currently stands. Therefore, we invite you to submit a revised version of the manuscript that addresses the points raised during the review process.

Please submit your revised manuscript by Jan 08 2026 11:59PM. If you will need more time than this to complete your revisions, please reply to this message or contact the journal office at plosone@plos.org. Please include the following items when submitting your revised manuscript:

We look forward to receiving your revised manuscript.

Kind regards,

Ivan Alejandro Pulido Tarquino, MSc

Academic Editor

PLOS ONE

Journal Requirements:

Additional Editor Comments:

Dear Dr. Tucker,

Thank you for your detailed responses and revised manuscript. We appreciate the improvements made, particularly in the Methods and Abstract.

However, after reviewing your replies alongside the reviewer reports and the revised text, some issues need to be addressed.

To proceed, please revise the manuscript and your response letter according to the points below.

We are still unable to view Figure 1. Please ensure that the file is correctly uploaded and that the figure is visible in the revised submission.For a better reading experience, please also upload a clean (tracked-changes–free) version of the manuscript alongside the marked-up version.

Thank you

Kind regards

Dr. Ivan Alejandro Pulido Tarquino

Academic Editor

Reviewers' comments:

Reviewer's Responses to Questions

**Comments to the Author**

1. If the authors have adequately addressed your comments raised in a previous round of review and you feel that this manuscript is now acceptable for publication, you may indicate that here to bypass the “Comments to the Author” section, enter your conflict of interest statement in the “Confidential to Editor” section, and submit your "Accept" recommendation.

Reviewer #1: All comments have been addressed

2. Is the manuscript technically sound, and do the data support the conclusions?

Reviewer #1: Partly

3. Has the statistical analysis been performed appropriately and rigorously? 

Reviewer #1: N/A

4. Have the authors made all data underlying the findings in their manuscript fully available?

Reviewer #1: Yes

5. Is the manuscript presented in an intelligible fashion and written in standard English?

Reviewer #1: Yes

6. Review Comments to the Author

Reviewer #1: revision is well done. Unfortunately I still cannot read Fig 1 with very important informations and i cannot find the themes and categories (see line 147/148 with error This has to be corrected

7. PLOS authors have the option to publish the peer review history of their article (what does this mean?). If published, this will include your full peer review and any attached files.

Reviewer #1: No

---

## [Author Response · Author response to Decision Letter 2]

2 Dec 2025

Dear Ivan Alejandro Pulido Tarquino,

Thank you for your comments on our revision. There were two minor points that were flagged:

1) Figure 1 not readable. Apologies that we uploaded a JPG that was difficult to read. We have used the SASS tool to create a TIF file. This is now uploaded and should make the themes easier to appreciate. The themes are also described in the text.

2) Clean version. We have uploaded a clean version of the manuscript.

Please let me know if you need anything further.

Yours sincerely,

Joseph D. Tucker

---

## [Decision Letter · Decision Letter 2]

6 Apr 2026

PONE-D-25-14440R2Online sexual health survey implementation: Practical lessons from a comparative, mixed-methods analysis of 30 countriesPLOS One

Dear Dr. Tucker,

Thank you for submitting your manuscript to PLOS ONE. After careful consideration, we feel that it has merit but does not fully meet PLOS ONE’s publication criteria as it currently stands. Therefore, we invite you to submit a revised version of the manuscript that addresses the points raised during the review process.

The manuscript has been evaluated by two reviewers, and their comments are available below.

The reviewers have raised a number of concerns that need attention. Could you please revise the manuscript to carefully address the concerns raised?

We look forward to receiving your revised manuscript.

Kind regards,

Johanna Pruller, Ph.D.

Senior Editor

PLOS One

Journal Requirements:

Reviewer's Responses to Questions

**Comments to the Author**

1. If the authors have adequately addressed your comments raised in a previous round of review and you feel that this manuscript is now acceptable for publication, you may indicate that here to bypass the “Comments to the Author” section, enter your conflict of interest statement in the “Confidential to Editor” section, and submit your "Accept" recommendation.

Reviewer #1: All comments have been addressed

Reviewer #2: (No Response)

2. Is the manuscript technically sound, and do the data support the conclusions?

Reviewer #1: (No Response)

Reviewer #2: Yes

3. Has the statistical analysis been performed appropriately and rigorously? 

Reviewer #1: (No Response)

Reviewer #2: Yes

4. Have the authors made all data underlying the findings in their manuscript fully available?

Reviewer #1: (No Response)

Reviewer #2: Yes

5. Is the manuscript presented in an intelligible fashion and written in standard English?

Reviewer #1: (No Response)

Reviewer #2: Yes

6. Review Comments to the Author

Reviewer #1: many thanks- i could read now fig 2 after downloading it separately and I have no further remarks. Interesting article

Reviewer #2: Thank you for the opportunity to review this manuscript. The authors have presented interesting mixed-methods data about several countries' experiences using online surveys to collect information about sexual health during the pandemic. I can see that comments from the previous reviewer have been addressed. I have just a few minor points to add:

1. The authors might reflect on the role of online surveys more specifically to sexual health. Did any countries report that people were more likely to take part in an online survey about sexual health because it could be viewed as more private? Overall, the paper focuses a lot on the online methodology, but less about how this interacts with the survey content, which can be quite sensitive. If these sorts of discussions did not come up, the authors might list this as a limitation or area of future research.

2. The authors may wish to reference, in the introduction or discussion, the Natsal-COVID study in the UK, which achieved a quasi-representative study based on comparisons to census data. Results and methods publications for this study are readily available. The authors discuss issues with representativeness in the included studies, but do not explain how representativeness was measured. This would be helpful to add. https://wellcomeopenresearch.org/articles/7-166

3. In lines 145-146, there seems to be an issue with the referencing software.

4. The authors may wish to more explicitly explain how their findings can inform future sexual health research post-pandemic, particularly in the context of survey fatigue.

7. PLOS authors have the option to publish the peer review history of their article (what does this mean?). If published, this will include your full peer review and any attached files.

Reviewer #1: **Yes:**Erika Baum

Reviewer #2: No

---

## [Author Response · Author response to Decision Letter 3]

20 Apr 2026

Dear Dr. Pruller,

Thank you for providing helpful comments on our manuscript. We have used them to strengthen the manuscript.

Find attached point-by-point responses. Please note line numbers correspond to the Tracked Changed document in red and the clean document in black.

Sincerely,

Hanna and Joe on behalf of the co-authors

---

## [Decision Letter · Decision Letter 3]

13 May 2026

Online sexual health survey implementation: Practical lessons from a comparative, mixed-methods analysis of 30 countries

PONE-D-25-14440R3

Dear Dr. Tucker,

We’re pleased to inform you that your manuscript has been judged scientifically suitable for publication and will be formally accepted for publication once it meets all outstanding technical requirements.

Kind regards,

Zypher Jude G. Regencia, Ph.D.

Academic Editor

PLOS One

Additional Editor Comments (optional):

Reviewers' comments:

Reviewer's Responses to Questions

**Comments to the Author**

1. If the authors have adequately addressed your comments raised in a previous round of review and you feel that this manuscript is now acceptable for publication, you may indicate that here to bypass the “Comments to the Author” section, enter your conflict of interest statement in the “Confidential to Editor” section, and submit your "Accept" recommendation.

Reviewer #1: All comments have been addressed

Reviewer #2: All comments have been addressed

2. Is the manuscript technically sound, and do the data support the conclusions?

Reviewer #1: Yes

Reviewer #2: (No Response)

3. Has the statistical analysis been performed appropriately and rigorously? 

Reviewer #1: Yes

Reviewer #2: (No Response)

4. Have the authors made all data underlying the findings in their manuscript fully available?

Reviewer #1: No

Reviewer #2: (No Response)

5. Is the manuscript presented in an intelligible fashion and written in standard English?

Reviewer #1: Yes

Reviewer #2: (No Response)

6. Review Comments to the Author

Reviewer #1: Where is reference for COVID natal study in literature list?

Could you please mark it adequately in text and literature list?

Reviewer #2: (No Response)

7. PLOS authors have the option to publish the peer review history of their article (what does this mean?). If published, this will include your full peer review and any attached files.

Reviewer #1: **Yes:**Erika Baum

Reviewer #2: No

---

## [Editor Report · Acceptance letter]

PONE-D-25-14440R3

PLOS One

Dear Dr. Tucker,

I'm pleased to inform you that your manuscript has been deemed suitable for publication in PLOS One. Congratulations! Your manuscript is now being handed over to our production team.

Kind regards,

on behalf of

Dr. Zypher Jude G. Regencia

Academic Editor

PLOS One